# Impact of improved stove intervention on infant acute respiratory infections: Results from a randomized trial in Northwest Ethiopia

**Habtamu Demelash Enyew**[1,2]*, **Abebe Beyene Hailu**[2], **Seid Tiku Mereta**[2]

**1** Debre Tabor University, College of Health Sciences, Department of Public Health, Debre Tabor, Ethiopia, **2** Jimma University, Institute of Health, Department of Environmental Health Science and Technology, Jimma, Ethiopia

* enyew29@gmail.com

## Abstract

### Background

Globally, millions of young children affected by acute respiratory infections every year, which is the leading cause of death and serious illness among children. Though strategies, including promoting improved stoves, have been implemented to combat this public health challenge, the effectiveness of local improved stoves introduced during pregnancy in reducing household air pollution and related respiratory illnesses remains limited.

### Method

Following the main trial randomization, 343 infants born to mothers in the study groups were followed for six months, with assessments of acute respiratory infections (ARI) occurring every two months. The stove intervention's impact was evaluated by comparing the acute respiratory infection incidence rate between the intervention and control groups. Respiratory illnesses were assessed using Integrated Management of Childhood Illness (IMCI) guidelines. The incidence rate ratio (IRR) was estimated using a marginal Poisson model fitted via Generalized Estimating Equations (GEE).

### Result

During the six-month follow-up period, a total of 43 infants (18 intervention, 25 control) experienced at least one acute respiratory infection (ARI) episode, resulting in a cumulative incidence of 12% (95% CI: 10, 14%). Although the intervention group consistently showed a reduction in ARI incidence rates compared to the control group (a 20% reduction), the adjusted Incidence Rate Ratio (IRR = 0.81; 95% CI: 0.56, 1.16; P = 0.252) was not statistically significant. A non-significant trend toward benefit was noted overall, with the subgroup analyses suggesting promising, non-significant reductions primarily among female infants and infants from larger families.

**Data availability statement:** All relevant data are within the paper and its Supporting Information files.

**Funding:** The author(s) received no specific funding for this work.

**Competing interests:** The authors have declared that no competing interests exist.

## Conclusion

The overall study did not find a statistically significant protective effect of the intervention stoves on ARI incidence. However, the observed non-significant trend suggests a potential protective effect. Further research with larger sample sizes and longer follow-up periods is warranted to confirm this potential benefit.

## Trial registration

The main trial was registered at the Pan African Clinical Trial Registry website under the code PACTR202111534227089, (https://pactr.samrc.ac.za/ (Identifier) on the registration date of (11/11/2021).

## Introduction

Infants mortality remains a pressing challenge globally with millions losing their lives in the first year of life [1] particularly in Sub-Saharan Africa where neonatal mortality rates remain high [1,2]. According to world health organization (WHO) report of 2020, Ethiopia is the 4th country next to India, Nigeria and Pakistan from the top 10 countries with the highest number of newborn deaths [1]. Acute respiratory infections (ARIs) including pneumonia consistently stands as one of the leading causes of morbidity and mortality in young children on a global scale [2,3] and responsible for between 1.9 and 2.2 million childhood deaths [4].

Early life exposures to household air pollution, can have lasting impacts on health and susceptibility to ARIs even later in life [5–8]. Burning biomass fuels is a known risk factor for ARIs [9–14]. Studies from around the world consistently show a link between household air pollution (HAP) exposure and increased ARI risks, especially pneumonia [9,13,15,16]. The main health-damaging pollutants in this context are fine particulate matter with aerodynamic diameters smaller than 2.5μm ($PM_{2.5}$) and carbon monoxide (CO) [17–19]. The burden of ARIs associated with HAP is disproportionately high in certain regions, including sub-Saharan Africa where reliance on solid biomass fuels remains high [20,21].

The substantial increase in health risk associated with HAP calls for immediate interventions [22–24] and distributing improved stoves has been identified as the best option for a safe transition to clean cooking in poor rural African settings, including Ethiopia [25]. Improved stoves are also aligned with several Sustainable Development Goals, including good health and well-being, gender equality, affordable and clean energy, and climate action [18,26,27]. Recognizing these benefits, the Ethiopian government has set a target of distributing improved stoves to 31 million households by 2030 [28]. However, the effectiveness of locally available improved stoves in reducing air pollutants and associated health impacts, particularly in terms of acute respiratory illnesses in young children, remains uncertain and lacks definitive quantifiable data [29]. Therefore, the primary goal of this study was to measure the impact of a locally manufactured improved cook stove intervention on PM2.5 exposure and the incidence of acute respiratory infections (ARI) among infants during the first six months of life.

## Methods

### Study setting and period

The study was conducted in the rural areas of South Gondar Zone, Northwest Ethiopia which is characterized by diverse climatic conditions with local community commonly using biomass fuels for cooking over open fires. Pregnant women were recruited from two distinct ecological zones (cold and temperate) to capture a diversity of characteristics known to influence cooking practices, including altitude, fuel types, population density, and socioeconomic conditions. The community is typical of many rural poor communities in the wider region, lending the findings reasonable generalizability and wide relevance in the region. Given the low prevalence of both tobacco smoking and vehicle emissions, kitchen smoke from the widespread use of traditional three-stone biomass stoves represents the primary source air pollution exposure in the study area [30,31].

### Study design

This study is a follow-up to a randomized controlled trial that investigated the impact of improved stoves on pregnancy outcomes. Infants born to mothers who participated in the original trial were followed up for 6 months to assess the occurrence of ARIs. Eligibility for the main trial was limited to pregnant women who were aged 18–38 years, carrying singleton fetus (not known twins), women with no known pre-existing health conditions and were in their first or second trimester of gestation (determined by the self-reported first day of the last menstrual period and via ultrasound-where available). Furthermore, participants had to be the primary cook in their household and exclusively utilize either a traditional biomass-fueled stove or a locally modified mud stove [30].

### Sample size

A follow-up of all live births from the parent trial was done. All infants born alive and with at least one fieldworker ARI surveillance visit were included in the analyses. Accordingly, 343 infants (176 in the intervention group and 167 in the control group) were followed until the age of 6 months. Since the sample size for this follow-up was fixed based on the live births from the parent trial, we employed a minimum detectable effect size calculation. Accordingly, the minimum detectable difference for this study is an Incidence Rate Ratio (IRR) of approximately 0.70, corresponding to a 30% reduction in the incidence of ARI.

### Intervention

This trial involved introducing the Mirt stove which is locally sourced improved stove with a chimney, to households in the intervention group. This intervention stove will be designated as the "improved stove" throughout this article. This intervention aimed to reduce HAP and potentially improve the health of new born infants. Because, traditional open-burning stove exposes families including infants to high levels of harmful particulate matters due to incomplete combustion and lack of proper ventilation [31].

The intervention stove is featured with a sealed combustion chamber, chimney, a clay baking pan and the fuel feeding door allows for the addition of fuel without interrupting the cooking process (Fig 1). Previous studies indicated that this improved stove even without chimney reduces fuelwood usage nearly by half (51%), carbon monoxide emissions by 92%, and particulate matter ($PM_{2.5}$) by 41% when compared to traditional three-stone stove [32,33]. The intervention stove was sold at a market price of 400 birr (equivalent to approximately $8.00 USD at the beginning of the study) and study participants received the stoves at a half subsidized rate.

### Randomization

Following informed consent and baseline assessments, participants were randomly allocated in a 1:1 ratio to either the intervention group (receiving an improved stove) or the control group (continuing to use a traditional three-stone stove)

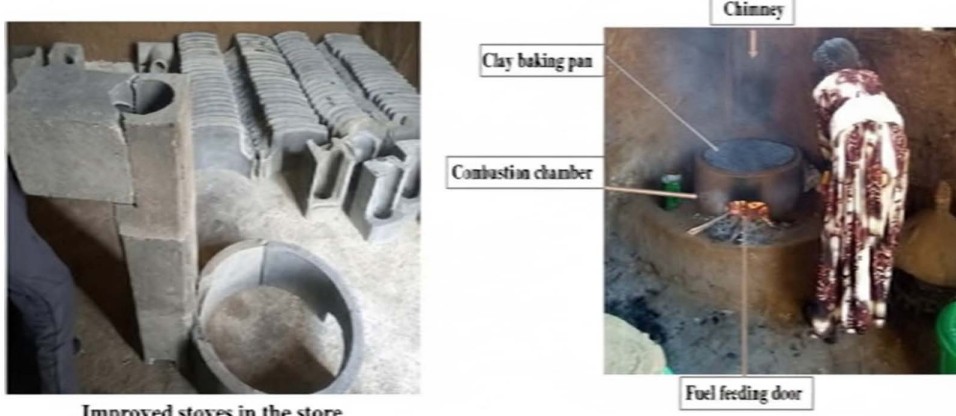

**Fig 1. Chimney fitted improved stove in the store before installation and in function after installation.**

(Fig 2). Infants subsequently born to these mothers were automatically assigned to the same study arm as their mother, as no separate randomization was conducted for the infants. The allocation sequence was independently generated by the biostatistician using an Excel random number generator. This simple randomization technique ensured known and unknown baseline confounding factors were evenly distributed between the arms to mitigate selection bias. Although the biostatistician was responsible for generating the sequence and assigning participants, blinding was considered unfeasible due to the inherent, visible nature of the stove installation.

## Outcome assessment

The incidence of ARIs among infants was assessed by trained field data collectors using the standard definitions and protocols outlined in the Integrated Management of Childhood Illness (IMCI) module [34]. An ARI episode was specifically defined as mother-reported cough, difficulty breathing, or runny/stuffy nose lasting on at least two consecutive days, with a minimum of two symptoms reported on at least one day, and at least one symptom present on each day. To mitigate recall bias, parents were rigorously instructed to contact data collectors, health extension workers, or local health development army leaders immediately upon observing these symptoms. Furthermore, data collectors were trained to maintain a neutral approach and emphasize accurate reporting of all symptoms (or lack thereof), ensuring consistency over time and mitigating reporting bias.

## Data collection

The baseline data for the parent trial was collected using structured questionnaire as detailed in the previous publication [30]. Trained fieldworkers visited each infant's home every two months throughout their six months of life. During the scheduled active surveillance, fieldworkers directly asked mothers if their infants had experienced fever, persistent cough, and wheezing/difficulty breathing in the past two months before the visits. The monitoring continued until each infant reached six months of age, at which point they were discharged from the study. This lasts from August 12, 2023 to January 15, 2024. In addition to the outcome variable, data were collected on the potential factors that may affect the occurrence of ARIs including vaccination status for age and exclusive breast feeding.

Efforts were made throughout the study to minimize the potential for reporting bias, particularly because the outcome (ARI episodes) relied on mother-reported data. We utilized a strict, multi-criterion definition of ARI to objectify the subjective reporting of symptoms (requiring two consecutive days with a specific symptom count). Data collection occurred

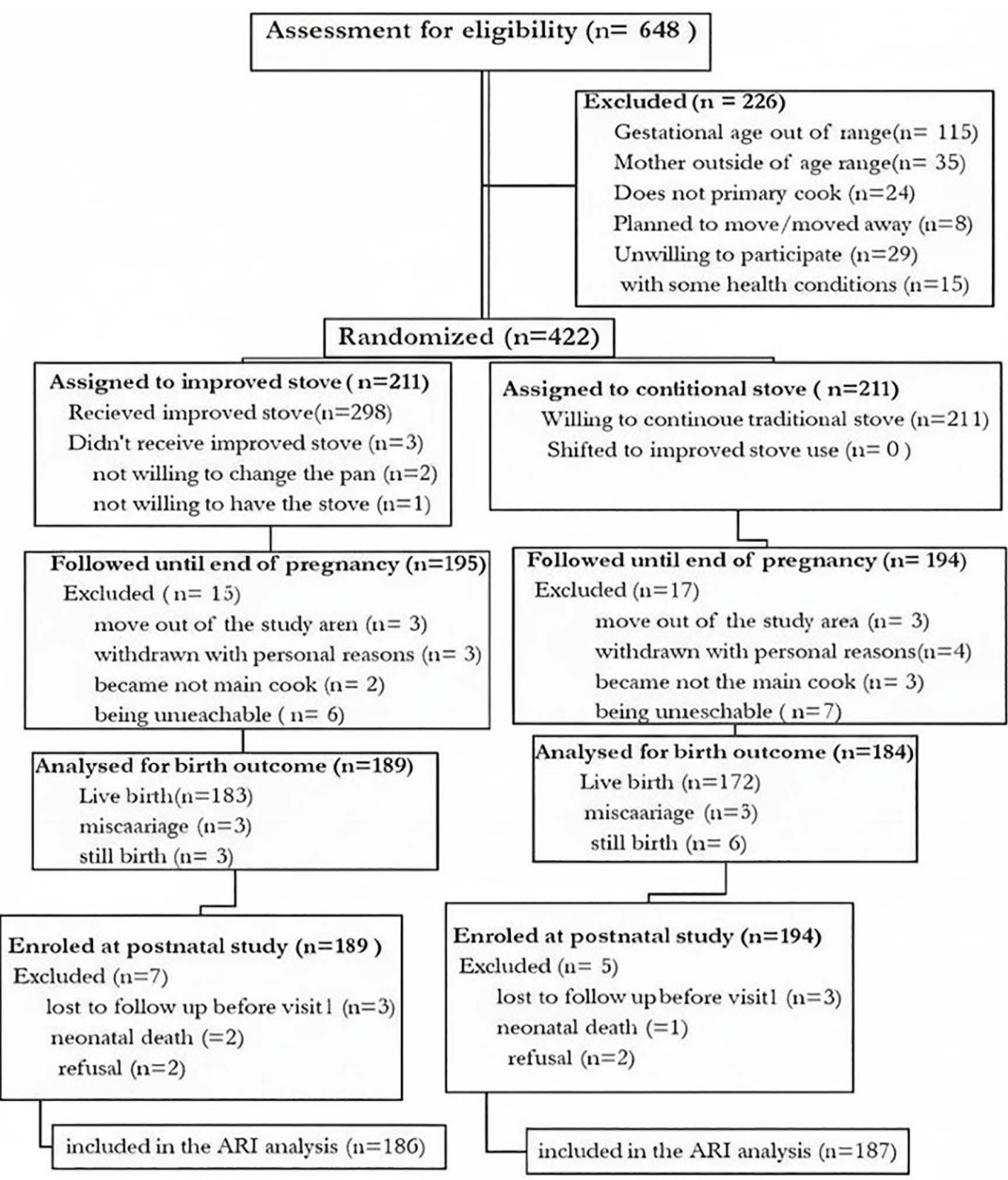

**Fig 2. Consolidated Standards of Reporting Trials (CONSORT) flow diagram.**

during routine, predetermined follow-up visits, rather than only when the infant was acutely ill. This systematic approach helped to mitigate recall bias and ensured that symptom reporting was captured consistently across all infants throughout the follow-up period, regardless of the perceived severity of any single episode.

## Statistical analysis

We excluded lost to follow-up before the first visit, experienced neonatal death, or refused to participate. For the primary analysis, we employed an available case approach by only considering the complete data available for each visit, as we

did not impute missing data. A baseline descriptive analysis was done to compare the intervention groups on important characteristics. Independent sample t-tests and chi-square statistical tests were used for continuous and categorical data respectively. Given the longitudinal design with repeated infant visits, we analyzed the count data for ARI episodes using a marginal Poisson regression model, estimated via Generalized Estimating Equations (GEE) using an Exchangeable Working Correlation Structure to account for the correlation arising from the repeated measurements taken within the same infant over the 6-month follow-up period. GEEs enhance this analysis by estimating how the risk of ARI changes over time, considering the potential correlation between repeated visits from the same infant. To calculate incidence rates (IR) and risk rate ratios (IRRs), we used infant-months at risk as the denominator. Since the benefits of the original randomization are diminished in this subsequent, non-randomized cohort, making an unadjusted primary analysis potentially susceptible to bias from remaining confounders. Therefore, adjustment was made to ensure our primary analyses are robust and appropriately address potential confounding in the absence of the protective benefits of the original randomization.

The incidence of ARIs was also assessed within subgroups which allows to explore whether the intervention's effectiveness (reduction in ARI) varies depending on certain characteristics. To assess potential variations in the intervention's effect, we categorized our infant population based on nine characteristics identified from previous studies to be potentially associated with ARI risk. These characteristics include family size, kitchen location, kitchen roof type, frequency of "Injera" baking per week, wealth index, infant sex, age, gestational age, exclusive breastfeeding practices and vaccination status for the age which are potentially related to ARI risk [35–37]. All statistical analyses were conducted using SPSS version 24 [38].

## Results

### The trial profile

From the total 422 pregnant women recruited and randomly assigned to either the intervention or control group in the main trial (Fig 2), three participants in the intervention group did not receive the improved stove and were excluded from the final analysis, leaving 419 pregnant women for follow-up. During the course of the study, 30 pregnant women became non-adherent and were also excluded and we documented 17 pregnancy losses. Finally, 372 birth outcomes were included in the pregnancy outcome analysis and 355 infants were enrolled in the postnatal study.

However, only 343 infants were ultimately included in the ARI analysis as 12 infants were not included due to loss to follow up before the first visit, neuronal death and withdrawal of consent that occurred between the initial postnatal enrollment and the final ARI outcome assessment.

### Pregnancy and infant characteristics

The baseline and infant characteristics of participants who remained in the study after exclusions was compared to ensure comparability between groups. The participants (as demonstrated in Table 1), from intervention and control groups were well-balanced on most relevant pregnancy and newborn characteristics, except the average maternal age at delivery between the intervention and control groups (28.6 and 29.3 years, respectively; p = .023) and infant vaccination completion for their age (intervention: 158 [86.4%] vs. control: 141 [82.5%]; p = .048). These differences might influence ARIs, highlighting the importance of including them in adjusted analyses to account for their potential role in the intervention effectiveness.

### Acute respiratory infection outcomes

A total of 343 infants (176 in the intervention group and 167 in the control group) were included in the follow-up cohort, being tracked every two months for the first six months of life with a total of 1,029 visits over the study period. During the

**Table 1. Pregnancy, delivery, and infant characteristics among mother-infant pairs stratified by Intervention arm in the South Gondar Zone, Northwest Ethiopia.**

| Characteristics | Overall (n = 355) | Intervention (n = 183) | Control (n = 172) | p-value* |
|---|---|---|---|---|
| Maternal age at delivery, mean± SD | 28.9±5.2 | 28.6±5.2 | 29.3±5.1 | .023 |
| Total family size, mean±SD | 4.8±1.6 | 4.7±1.6 | 4.9±1.6 | .133 |
| ANC complete (at least 4 visit), n(%) | | | | |
| Yes | 97 | 49 (26.7) | 48 (27.9) | .313 |
| No | 255 | 127 (72.2) | 116 (70.6) | |
| Missed | 3 | 1 | 2 | |
| Current pregnancy order, n (%) prim gravida | 23 | 12(6.6) | 11 (6.4) | .514 |
| Multigravida | 332 | 171 (93.4) | 161 (93.6) | |
| Infant sex, n (%) | | | | |
| Male | 174 | 90 (48.9) | 84 (49.5) | .446 |
| Female | 181 | 94 (51.1) | 87 (50.5) | |
| Mean birth weight, in grams, ±SD | 3050±497 | 3045±491 | 3043±499 | .944 |
| Range | 1315 - 4200 | 1700 - 4200 | 1315 - 4162 | |
| Gestational age at birth, weeks | 39.4±1.9 | 39.3±1.9 | 39.4±1.8 | .640 |
| Range | 32-42 | 32-42 | 32-42 | |
| Low birth weight, n (%) | | | | |
| Yes (<2500 grams) | 26 | 13 (8.2) | 13 (7.7) | |
| No (≥2500 grams) | 304 | 159 (92.3) | 145 (91.8) | .433 |
| Preterm birth, n (%) | | | | |
| Yes (< 37 weeks) | 18 | 10 (5.6) | 8 (4.9) | |
| No (≥ 37 weeks) | 298 | 154 (84.0) | 144 (83.7) | .739 |
| Pentavalent and PCV | | | | |
| Completed | 299 | 158(86.4) | 141(82.5) | .048 |
| Not completed | 43 | 18(9.8) | 25(14.6) | |
| Exclusive breast feeding | | | | |
| Yes | 178 | 96(58.0) | 82(47.2) | .669 |
| No | 165 | 69(42.0) | 96(52.8) | |
| Place of delivery | | | | |
| Home | 35 | 16(8.7) | 19(10.9) | .378 |
| Institute | 308 | 160(87.5) | 148(86.2) | |
| Mode of delivery | | | | |
| Svd | 258 | 132 (72.2) | 126 (73.2) | .243 |
| Instrumental assisted | 16 | 7 (3.6) | 8 (4.1) | |
| Episiotomy | 15 | 10 (5.6) | 5 (3.1) | |
| Asset index | | | | |
| Poor | 120 | 57 (31.0) | 63 (36.8) | .114 |
| Middle | 120 | 63 (34.7) | 57 (36.8) | |
| Rich | 115 | 63 (34.3) | 52(29.9) | |
| Drinking water source | | | | |
| Improved | 179 | 95 (51.9) | 84 (49.1) | .390 |
| Unimproved | 175 | 88 (48.1) | 87 (50.9) | |
| Toilet types | | | | |
| Improved | 92 | 44 (23.5) | 48 (27.6) | .139 |
| Unimproved | 163 | 140 (76.5) | 123 (72.4) | |

*Compression for continuous variables (maternal age, birth weight and gestational age) were done using independent sample t-test whereas for all categorical variables, chi-square test was done.

follow period, 42 episodes of ARIs were identified from 1,854 infant-months of observation. Recurrent ARI was relatively rare, with only 12 infants (4 from the intervention group and 8 from the control group) experiencing two ARI episodes throughout the follow-up period. The cumulative incidence of ARI was calculated as the proportion of infants who experienced at least one episode of ARI during the 6-month period. This was 11.8% (95% CI: 10, 14%) for the entire cohort. The cumulative incidence of ARI in the intervention and control arms was 10.7% (95% CI: 8%, 13%) and 13.0% (95% CI: 10%, 16%), respectively.

The proportion of infants with at least one ARI episode demonstrated a notable increase over the follow-up period for both groups. At the first visit, 14 infants (8.2%, 95% CI: 4, 12%) in the intervention group and 17 infants (9.9%, 95%CI: 5, 14%) in the control group had experienced an ARI episode. This proportion increased to 27 infants (15.9%, 95% CI: 10.4, 21.3%) and 27 infants (16.2%, 95% CI: 10.5, 21.8%) for the intervention and control arms, respectively, at the third visit. This indicates that while the number of ARI episodes increased with the age of the infants in both groups, the overall increase in incidence rates was comparable between the intervention and control arms throughout the follow-up period (Fig 3).

The average rate of new ARI episodes was calculated over the six-month follow-up period. Across all participants, the overall incidence rate (IR) was 5.67 ARI episodes (95% CI: 4.45, 7.07) per 100 infant-months (meaning nearly six ARI cases per 100 infants followed for one month). The control group exhibited a higher overall IR of 7.40 episodes (95% CI: 5.74, 9.05) per 100 infant-months compared to 6.16 episodes (95%CI: 4.67, 7.64) per 100 infant-months in the intervention group.

Both arms demonstrated a similar age-related increase, with incidence rates peaking at the final visit (6 months of age). In the control group, the rate increased from 5.45 episodes (95% CI: 2.96, 7.93) at the first visit to 8.72 episodes (95% CI: 5.67, 11.77) per 100 infant-months at the final visit. The intervention group also experienced a sharp increase, with rates rising from 4.48 episodes (95% CI: 2.29, 6.68) to 8.42 episodes (95% CI: 5.52, 11.32) per 100 infant-months at the final visit.

## Estimating the effect of the intervention

The primary analysis, using a marginal Poisson model estimated via GEE, indicated a 20% lower rate of ARI episodes in the intervention group compared to the control group, effectively accounting for the correlation of repeated measurements within infants. But, the resulting incidence rate ratio was not statistically significant (IRR = 0.80; 95% CI: 0.56, 1.15; P = 0.238) (Table 2). For comparison, a standard Poisson regression model (which does not account for within-subject

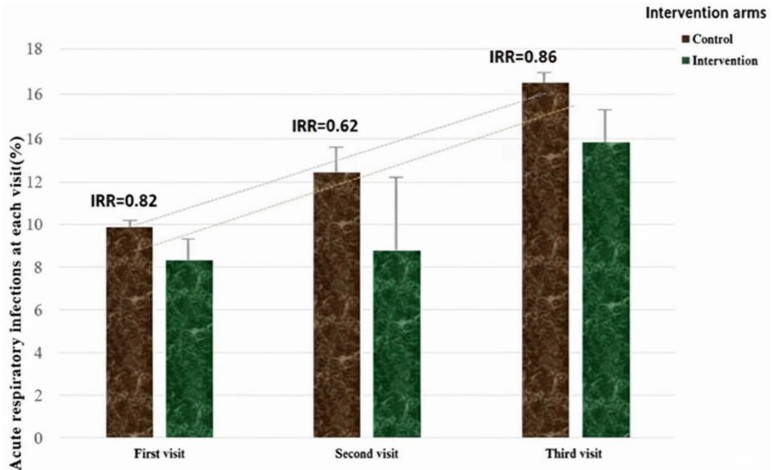

**Fig 3. The longitudinal incidence rates of ARIs episodes at each visit.**

**Table 2. Effect of the intervention stove against fieldworkers assessed ARIs among infants in south Gondar zone, northwest Ethiopia.**

| Characteristics | ARI Episodes | | Unadjusted IRR (95% CI) | P Value* | Adjusted IRR (95% CI) | P Value** |
|---|---|---|---|---|---|---|
| | Yes | No | | | | |
| Intervention arms | | | | | | |
| Intervention group | 20 | 163 | 0.80 [0.56 - 1.15) | 0.238 | 0.81 [0.56-1.16] | .252 |
| Control group | 23 | 149 | 1 | – | 1 | |
| **Subgroup analysis** | | | | | | |
| Infant sex | | | | | | |
| Female | 18 | 162 | 0.58[0.33 - 1.03] | .062 | 0.59[0.34–1.04] | .071 |
| Male | 24 | 151 | 1.04[0.63 - 1.71] | .887 | 0.95[0.63 - 1.44] | .894 |
| Gestational age at birth | | | | | | |
| Preterm (< 37 weeks) | 4 | 15 | 0.61[0.16 - 2.34] | .470 | 0.55[0.14 - 2.21] | .400 |
| Term (≥ 37 weeks) | 35 | 263 | 0.79[0.53 - 1.19] | .267 | 0.80[0.53 - 1.20] | .278 |
| Exclusive breast feed | | | | | | |
| Yes | 18 | 163 | 1.01[0.56 - 1.80] | .977 | 1.00[0.56 - 1.80] | .989 |
| No | 24 | 138 | 0.73[0.45 - 1.17] | .194 | 0.74[0.46 - 1.19] | .218 |
| Pentavalent and PCV | | | | | | |
| Completed | 37 | 163 | 0.84[0.57 - 1.25] | .398 | 0.85[0.57 - 1.26] | .417 |
| Not completed | 5 | 38 | 0.59[0.21 - 1.64] | .314 | 0.58[0.21 - 1.61] | .297 |
| Family size | | | | | | |
| <5 occupants | 8 | 94 | 1.06 [0.66 - 1.69] | .814 | 1.03[0.69 - 1.53] | .793 |
| ≥5 occupants | 34 | 206 | 0.54[0.30 - 0.98] | .045 | 0.48 [0.29- 0.81] | .079 |
| Kitchen roof type | | | | | | |
| Thatched roof | 10 | 91 | 0.80[0.37 - 1.71] | .569 | 0.80[0.37 - 1.71] | .566 |
| Corrugated iron sheet | 32 | 222 | 0.80[0.53 - 1.22] | .304 | 0.81[0.53 - 1.23] | .327 |
| Kitchen location | | | | | | |
| Separated | 29 | 216 | 0.80 [0.52 - 1.23] | .316 | 0.81[0.53 - 1.25] | .351 |
| Attached | 12 | 94 | 0.92 [0.46 - 1.82 | .808 | 0.91[0.46 - 1.81] | .799 |
| Injera baking per week | | | | | | |
| Once | 7 | 56 | 0.82[0.35 - 1.92] | .644 | 0.85[0.36 - 2.04] | .728 |
| Twice and more | 35 | 257 | .80[0.52 - 1.20] | .280 | 0.80[0.54 - 1.19] | .281 |
| Asset index | | | | | | |
| Poor | 15 | 104 | 0.67[0.35 - 1.27] | .225 | 0.67[0.35 - 1.27] | .222 |
| Middle | 11 | 109 | 1.78[0.85 - 3.76] | .128 | 1.71[0.82 - 3.59] | .154 |
| Rich | 15 | 70 | 0.57[0.31 - 1.09] | .088 | 0.61[0.32 - 1.15] | .129 |

*Differences for categorical variables come from Poisson regressions with robust estimates. Estimates shown represent risk ratios and 95% CIs as compared with the control group. Differences replicate unadjusted models (*) and additionally adjust for covariates (**).

Abbreviations: ARI, acute respiratory infection; IRR, incidence rate ratio.

correlation) yielded a very nearly similar result, showing an 18% reduction (IRR = 0.82; 95% CI: 0.59, 1.14; P = 0.250). While these reductions were not statistically significant, both incidence rate ratios (IRRs) suggest a protective effect of the intervention (improved stoves) on the incidence rate of ARI among infants.

GEE can model the probability of having ARI at each visit while accounting for correlations between repeated measures on the same infant. While this reduction was not statistically significant, the point estimates suggest a potentially protective effect of the improved stove, particularly at earlier time points. The intervention appeared to provide a greater protective

effect at the second visit (4 months of age) (IRR = 0.62; 95% CI: 0.32, 1.22; P = .166) and the lowest protective effect at the third visit (6 month of age) (IRR = 0.96 (95%CI: 0.55, 1.72; P = .918) (Fig 3). Despite not reaching statistical significance, the adjusted GEE model showed a promising 19% reduction in ARI incidence among infants exposed to improved stoves compared to the control group (IRR = 0.81; 95% CI: 0.56,1.16; P = .252) after controlling maternal age and infant vaccination completion for their age.

From our subgroup analysis, being female infant and living in households with five or more occupants revealed potential benefits from the intervention. However, the reduction in ARI incidence for females in the intervention group (IRR = 0.59; 95% CI: 0.34, 1.04; P = .071) was not statistically significant. In contrast, infants living in households with five or more occupants in the intervention group had a 46% reduction in ARI incidence (IRR = 0.54; 95% CI: 0.30, 0.98, p = .045) compared to the control group. While the unadjusted analysis suggested a statistically significant effect, this effect became less pronounced after adjusting for maternal age at delivery and infant vaccination completion (IRR = 0.48; 95% CI: 0.29, 0.81; p = .079). Despite observing trends towards reduced ARI incidence in some subgroups, these findings did not reach statistical significance (Table 2).

## Discussion

A growing body of evidence links household air pollution from biomass fuel use, resulting in high indoor air pollution levels, with ARIs in rural Ethiopia [39–42]. To address concerns about HAP and investigate the potential protective effect of improved stoves, we evaluated whether introducing chimney-fitted Mirt stoves during pregnancy and continuing their use until infants reached 6 months of age could reduce ARI incidence. We employed a randomized controlled trial to compare the incidence of ARIs in infants living in households with intervention stoves versus those with traditional stoves in typical rural Ethiopian homes [30,31].

Despite the lower overall burden of acute respiratory infection (ARI) observed in the intervention group during the first six months of life, the stove intervention did not demonstrate a statistically significant reduction in ARI incidence. This outcome presents a mixed picture compared to related literature. It partially aligns with the KidsAir study in rural U.S. households [43], which also reported no meaningful difference in lower respiratory tract infections following air filtration intervention. Conversely, this result differs from a cluster-randomized controlled trial in Rwanda [44], which documented a statistically significant decrease in ARI among children under five years old after introducing rocket-style cook stoves. This variation is expected, as the effectiveness of the intervention is highly dependent on local context, including fuel type, cooking methods, meal type, and kitchen style.

The overall ARI incidence observed in this study falls slightly below the rate reported in a comparable study conducted in Ethiopia [35]. This potential difference might be attributed to the extended duration of our follow-up (1-year), variations in the age range of the study children (under-five children), and differences in sample size. Conversely, the ARI incidence rate in our study is higher than the report from Mexico [45]. These observed differences could potentially be attributed to the nature of the specific intervention stoves used, variations in sample size, the extended duration of follow-up, and differences in the age range of study children, factors known to influence ARI outcomes [46–48]. Beyond the health implications, addressing air pollution offers broader benefits, including driving economic growth and tackling climate change [49].

It was also observed that the magnitude of ARI incidence rose with the age of infants which aligns with the reported highest incidence rate for ARI and pneumonia occurring in the 9–12 months age group, compared to the lowest rate in infants under 3 months [50]. The worldwide pattern also revealed that the highest number of new cases occurred within the 1–4 years age groups [51]. The age group with the highest incidence of ARI is between 6 months and 2 years old, while the highest mortality rate from ARI occurs in infants under 1 year old [52]. The association between child age and ARIs was reported from additional literatures [53–55]. This trend may be due to cultural practices such as mothers breastfeeding or carrying children while cooking, which could inadvertently expose them to high levels of HAP [56]. Moreover,

as infants gain strength, they tend to spend more time near their mothers and engage in play with their younger siblings, leading to an increased exposure to air pollution [49,57].

As confirmed by both unadjusted and adjusted incidence risk ratio, the GEE model did not reveal a statistically significant reduction in ARI incidences attributable to the intervention stove. Previously reported trials in Malawi [58], Gahanna [59] Ethiopia [32,35], Guatemala [60] and Nepal [61] also found no statistically significant evidences that stove and fuel interventions reduced the risk of respiratory infections in young children. Despite established links between household air pollution and respiratory problems [10,14,62–64], the effectiveness of improved stoves in lowering exposures and ARI risk remains debatable, with research producing a range of inconclusive findings [35,44,58,65,66].

Insufficient reductions in HAP levels could explain the lack of observed improvements in respiratory health outcomes [67] as the concentrations of the pollutants were well above both the WHO guideline values from previous studies [68–70]. Our study's earlier findings revealed concerning levels of $PM_{2.5}$ in both intervention (104 µg/m3) and control (449 µg/m3) household kitchens after intervention [71]. These levels significantly exceed the WHO's recommended safe limit of 15 µg/m3 for a 24-hour period [72]. Though low adoption or stove stacking behaviors were not observed in our study, limitations reported in previous intervention studies, such as insufficient adoption or stove stacking may have contributed to the absence of a notable effect on child respiratory health [29,66,73,74]. Inconsistent use of the intervention, due to their complex maintenance and repair needs was also mentioned as one explanation for the limited effectiveness [67,75].

While our study found no significant impacts on ARI incidence, stove interventions in Rwanda [44] and fuel intervention in Nigeria [76] yielded protective impacts for respiratory health problems. Statistically significant reductions in persistent cough, wheeze and burn injury was also reported from rural Nepal [61] and a significant association was reported between Patsari stove intervention and a reduction of acute respiratory symptoms in rural Mexico [77]. It was also reported that using solid fuels for cooking without a chimney significantly increases the risk of acute lower respiratory infection (ALRI) among African children [78]. Children under five in Kenyan homes using traditional three-stone stoves experienced significantly higher rates of ARIs compared to those with improved stoves [79]. A systematic review found that improved cook stoves showed a reduction in respiratory and ocular symptoms among women, but no detectable impact on child health [80]. In Ethiopia, it was reported that stove intervention is associated with improvement in height-for-age of young children compared with a control group [32].

In our subgroup analysis, female infants and infants living in households with five and more occupants benefitted relatively more from the stove intervention. This is likely because larger family sizes can mean more exposure to cooking smoke [81] and a higher risk of respiratory infections for children, so the improved air quality from the intervention may have had a bigger effect in larger households [82,83]. However, the association between newborn sex and ARIs remains complex. For instance, there were findings indicating a higher incidence of ARIs among female children [37,40,84,85]. Similarly, studies from Bangladesh [86] and a more detailed study from Ghana [84] showed that female infants are particularly susceptible to respiratory infections. However, contradicted results have also been documented, suggesting that male fetuses may be more vulnerable to the effects of exposure to HAP in utero in Gahanna [87] and in Bangladesh [88,89]. Additionally, it was noted that females had a lower risk of hospitalization due to particulate matter compared to males [90]. The global mortality rate from lower respiratory infections was marginally higher in males [51].

The stove intervention, while a necessary step in addressing household air pollution, did not achieve the desired pollution reduction required to meet WHO air quality standards, leading to persistently high exposure levels that likely explain the lack of a statistically significant health impact on ARI incidence [35,65]. The persistently high pollutant levels underscore the practical difficulty of deploying improved stoves in rural areas, where factors like stove stacking, poor ventilation, and cultural cooking practices collectively undermine intervention effectiveness [91]. The findings underscore the fact that promoting improved stove is necessary but often insufficient [80]. Because, effective reduction of ARI requires a multifaceted approach that addresses behavioral change, improves housing ventilation, and ensures high and consistent usage of the improved stoves [92]. Therefore, future interventions must incorporate robust behavior change campaigns and

engineering solutions, like chimney maintenance or passive ventilation upgrades, to achieve the sustained, low-exposure environment necessary for tangible respiratory health benefits [93].

## Limitations of the study

The study may have been underpowered to detect smaller, yet clinically relevant, differences in ARI incidence, especially given the observed incidence rate ratio (IRR) values close to the null. It is also acknowledged that the sample size and the relatively short follow-up duration (six months) specifically limit the power to fully capture the long-term or cumulative effects of reduced household air pollution exposure. The findings are context-specific, influenced by the unique Ethiopian cultural practices related to food preparation (Injera baking), the specific design of the improved stove used, and the traditional kitchen configurations prevalent in the study setting. These local factors, which are often non-replicable in other geographic areas, should be considered when applying the results to different rural or resource-limited settings.

## Conclusion

In this study, the effectiveness of improved stove intervention in reducing acute respiratory infections in infants living in rural Ethiopia remains inconclusive. While the intervention did not yield statistically significant results, we observed promising trends suggesting a potential protective effect of the stoves on ARI incidence. Further research with larger sample sizes and longer follow-up periods is warranted to confirm these trends and elucidate the underlying mechanisms.

## Ethics approval

This study was ethically approved by Jimma University's institutional review boards (Ref No: IHRPGD/538/2021). All participating mothers provided written or verbal informed consent, depending on their literacy level and parental consent was obtained for participating infants as per the trial's standards. One copy of the signed consent form was given for each participant. Infants who were too ill to be treated at home were referred to the nearest health center. Official letters of cooperation were issued to the South Gondar Zone health department and relevant district health offices, securing permission to conduct the study. Throughout the research, participants were assured of their right to withdraw from interviews or the study altogether. Additionally, all information provided by participants was kept confidential and used solely for the study's stated objectives. Potential benefits of participation were also explained to all involved. The research was not imposing any risk to the participant as well as to the whole family

## Supporting information

**S1 File. Ari data.**
(SAV)

## Author contributions

**Conceptualization:** Habtamu Demelash Enyew, Abebe Beyene Hailu, Seid Tiku Mereta.

**Data curation:** Habtamu Demelash Enyew, Abebe Beyene Hailu, Seid Tiku Mereta.

**Formal analysis:** Habtamu Demelash Enyew, Abebe Beyene Hailu.

**Investigation:** Habtamu Demelash Enyew, Seid Tiku Mereta.

**Methodology:** Habtamu Demelash Enyew, Abebe Beyene Hailu, Seid Tiku Mereta.

**Project administration:** Habtamu Demelash Enyew.

**Validation:** Seid Tiku Mereta.

**Visualization:** Habtamu Demelash Enyew, Abebe Beyene Hailu, Seid Tiku Mereta.

**Writing – original draft:** Habtamu Demelash Enyew, Abebe Beyene Hailu, Seid Tiku Mereta.

**Writing – review & editing:** Habtamu Demelash Enyew, Abebe Beyene Hailu, Seid Tiku Mereta.

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
