## [Decision Letter · Decision Letter 0]

25 Sep 2025

Dear Dr. Demelash,

Thank you for submitting your manuscript to PLOS ONE. After careful consideration, we feel that it has merit but does not fully meet PLOS ONE’s publication criteria as it currently stands. Therefore, we invite you to submit a revised version of the manuscript that addresses the points raised during the review process.



We look forward to receiving your revised manuscript.

Kind regards,

Jennifer Tucker, PhD

Staff Editor

PLOS ONE

Journal Requirements:

“This study was financially supported by Jimma and Debre Tabor Universities”

3. Please include a copy of Table 1 & 2which you refer to in your text on page 6 & 8.

4. Please include a caption for figure 4.

5. Please ensure that you refer to Figure 4 in your text as, if accepted, production will need this reference to link the reader to the figure

Reviewers' comments:

Reviewer's Responses to Questions

**Comments to the Author**

1. Is the manuscript technically sound, and do the data support the conclusions?

Reviewer #1: No

Reviewer #2: Yes

Reviewer #3: Yes

Reviewer #4: Partly

Reviewer #5: No

2. Has the statistical analysis been performed appropriately and rigorously?

Reviewer #1: No

Reviewer #2: Yes

Reviewer #3: Yes

Reviewer #4: Yes

Reviewer #5: No

3. Have the authors made all data underlying the findings in their manuscript fully available?

Reviewer #1: Yes

Reviewer #2: Yes

Reviewer #3: Yes

Reviewer #4: Yes

Reviewer #5: No

4. Is the manuscript presented in an intelligible fashion and written in standard English?

Reviewer #1: Yes

Reviewer #2: Yes

Reviewer #3: Yes

Reviewer #4: Yes

Reviewer #5: No

Reviewer #1: This paper is a follow up study to a RCT that examined the impact of betters stoves on pregnancy outcomes. In this work, the authors follow the live births resulting from the original study for 6 months.

A number of clarifications and improvements are necessary.

1. Given that the sample size is fixed, please present what detectable difference was possible for given alpha and power. Please also account for clustering in this calculation.

2. Please describe the details of randomization.

3. When did the neonatal deaths occur? These should not be excluded. Was the cause of death respiratory infection?

4. What correlation structure was used for GEE? Were robust standard errors used? Please report these details in the statistical analysis section. Generally more details are needed. Please write out the model completely for me - including what the outcome looks like.

5. How was time parametrized in the model?

6. Were the subgroups defined a priori? THis is described in lins 216-223. Please move these lines to the methods.

7. THere is no ITT population. The mothers were randomized and some of these did not participate. THis is some type of completers analysis - as these were the mothers who completed the first study, and agreed to the follow up study. Given that the benefits of randomization do not apply here, the primary analyses should be adjusted for important confounders. Please describe this in the statistical analysis section, and remove references to the ITT analysis.

8. Were there any twins? how were these handled in the analysis?

9. Line 172 - 355 infants are referred to but there were only 343 live births. Please clarify.

10. Line 175 - what is a longitudinal prevalence? Please describe how this was estimated.

11.The statistical analysis section describe incidence rate of ARI, but in the results lines 173-175 prevalence is used. Please clarify.

12. It seems from the data collection that exact time of respiratory infection was collected, but in the analysis, only three time points were used? Please clarify.

13.Line 199. "GEE model" GEE is not a model. You are using a marginal Poisson model estimated via GEE.

14. Lines 199-201 results from a marginal model estimated via GEE and a Poisson model are described. In the stats section, it seemed like these were the same thing. Please clarify.

15. Subgroup analyses: In addition to presenting the subgroup specific effects, please include a p value that tests whether the effects differ in the subgroups (e.g. males vs. females).

16.Figure 2 would be more effective as a table with CIs for the IRRs and also showing crude and adjusted IRRs.

17. Line 294 - results described do not match that presented in the Figures - female or male infants? Same for abstract.

18. Many results are described in the text of the results section. Please make sure these are also included in tables.

19. There is no Table 1 or 2 - despite being referred to in the text.

20. Could you also present results that consider time as a continuous covariate? with a term for intervention and the interaction between time and interaction?

21. A sensitivity analysis that considers time to first respiratory infection, with time zero being birth and censoring occurring at the end of the study, adjusting for important confounders, using Cox PH regression would be a nice addition.

Reviewer #2: This randomized controlled trial examined whether providing locally made, chimney-fitted improved stoves (Mirt stoves) to pregnant women in rural Ethiopia could reduce the incidence of acute respiratory infections (ARIs) in their infants during the first six months of life. A total of 343 infants were monitored, with respiratory symptoms assessed every two months. The results showed that 12% of infants experienced at least one ARI episode, with a slightly lower incidence in the intervention group (10.7%) compared to the control group (13%). However, this difference was not statistically significant. The intervention group also showed a 19-20% lower risk of ARI, but again, this reduction did not reach statistical significance.

Subgroup analyses suggested possible benefits for female infants and those from larger families, but these trends were also not statistically significant. Importantly, indoor air pollution levels remained much higher than World Health Organization guidelines in both groups, which may have limited the health impact of the stove intervention. These findings align with other research in similar settings, where improved stoves often reduce household air pollution but do not consistently lower ARI rates in young children. The study concludes that while improved stoves may have some protective effect, larger and longer-term studies are needed to confirm these trends and better understand their impact on child respiratory health.

I appreciate the important contribution your study makes to understanding the impact of improved stoves on child respiratory health. I have several suggestions that I believe could strengthen your manuscript:

Clarify and Streamline the Abstract:

I recommend stating the primary outcome and key findings more explicitly in the abstract, for example, noting that no statistically significant reduction in ARI incidence was observed, but a non-significant trend toward benefit was noted. I also suggest shortening and focusing the abstract to better highlight the study design, main results, and implications.

Improve Reporting of Statistical Results:

I would find it helpful if you reported absolute numbers alongside percentages for key outcomes (e.g., "43 infants [12%] experienced at least one ARI episode"). Additionally, clarifying the meaning of IRR, CI, and p-values in lay terms would make the results more accessible. Providing effect sizes and confidence intervals for subgroup analyses in a concise table or summary would also be valuable.

Enhance Methodological Detail:

I suggest briefly explaining the rationale for using Generalized Estimating Equations (GEE), as some readers may be unfamiliar with this method. Please clarify the definition of ARI, specifying which symptoms were included and how they were assessed. It would also strengthen the manuscript to describe how missing data, dropouts, and losses to follow-up were handled in your analysis.

Address Potential Confounders and Bias:

I encourage you to explicitly discuss how differences in maternal age and vaccination status were controlled for in adjusted analyses. Please also mention any steps taken to minimize reporting bias in symptom reporting by mothers.

Strengthen the Discussion:

I would appreciate a more direct comparison of your findings with similar studies (such as those from KidsAir, Rwanda, and Mexico), including possible reasons for similarities or differences. Discussing the implications of persistently high PM2.5 levels despite the intervention, as well as the public health relevance and practical challenges of stove interventions in rural settings, would add valuable context.

Improve Figures and Tables:

Please ensure that all figures and tables are clearly labeled and referenced in the text. I also suggest adding a summary table of key outcomes (incidence, IRR, CI, p-value) for quick reference.

Refine Language and Structure:

I recommend editing the manuscript for clarity and conciseness, especially in the introduction and discussion sections. Removing repetition and using consistent terminology (e.g., always referring to the improved stove as "Mirt stove" or "improved stove") would enhance readability.

Address Limitations More Explicitly:

I believe it would be helpful to more clearly acknowledge the study's limited power to detect small differences due to sample size and follow-up duration. Discussing potential generalizability issues, such as unique features of the Ethiopian setting or cultural practices, would also be beneficial.

By addressing these points, I believe your manuscript will be clearer, more rigorous, and more compelling for both scientific and policy audiences. Thank you for your valuable work in this important area.

Reviewer #3: Dear Authors,

Greetings.

Your article PONE-D-25-24436 is well-written, and the scientific concept is strong. However, it requires further investigation with a larger sample size and a longer follow-up period. Additionally, the levels of PM2.5 in both the intervention (104 μg/m³) and control (449 μg/m³) household kitchens after the intervention significantly exceed the World Health Organization's recommended safe limits. This is an important point to consider when comparing with other studies.

The similarity report, which is attached, shows a similarity index of 33%, and I believe there is room for improvement in this area. Furthermore, the acronym HAP is first mentioned in line 74 without prior explanation. Additionally, in line 288 ARI, it is mistakenly referred to as ALRI.

All the best,  

Reviewer

Reviewer #4: Line 71: HAP is not defined before

Line 83-85: I understood the gap. But stating the goal of this study is missing. Few sentences could be added to explain the goal and objectives of this research.

Line 92: citations [30, 31]. I would prefer a summary or gist from those articles. Simply mentioning to check the articles may fail to describe the intention.

Line 97: Provide a little bit summary from that study.

Line 110: Picture resolution is poor

Line 110: Initially Mirt was written in italic form in line 103 but it is no longer italic in line 110

Line 139: Have any other studies used this method to explore similar or relevant topics? If yes, there should be some examples.

Line 139: Why did you use Poisson regression with generalized estimating equations? Explanation is missing. Why not other methods? What are most commonly used methods? Why this one differs from all other methods?

Line 143: Rephrase

Line 144: Why the mentioned variables are important to study?

Line 156-157: Check spacing

Line 257: citation bracket is in red color

Line 262-263: spacing between paragraphs

Line 293-294: spacing

Reviewer #5: Thank for giving me the opportunity to review this paper. I have the following comments and suggestions.

Sample Size:

1. No power calculation (sample size) for the study – the authors just mentioned that they followed 343 infants. So the study could be underpowered or overpowered.

2.

Statistical Analysis:

1. It is not clear what type of correlation structure was used in GEE.

Table 1 (baseline data), Table 2 (results table) mentioned in the text, but I didn’t see any such table. It is not possible to complete the review.

**Do you want your identity to be public for this peer review?** For information about this choice, including consent withdrawal, please see our Privacy Policy

Reviewer #1: No

Reviewer #2: No

Reviewer #3: **Yes: ** Beisan A. Mohammad

Reviewer #4: No

Reviewer #5: No

---

## [Author Response · Author response to Decision Letter 1]

2 Dec 2025

We appreciate the reviewers’ time in making detailed constructive comments and suggestions which have greatly helped to strengthen this manuscript. Below is our point-by-point response to each comment. We hope the revised manuscript captures better the study’s rigor, improves the quality of the writing and content, and appeals to a broader audience.

---

## [Decision Letter · Decision Letter 1]

23 Dec 2025

Impact of Improved Stove Intervention on Infant Acute Respiratory Infections: Results from a Randomized Trial in Northwest Ethiopia

PONE-D-25-24436R1

Dear Dr. Demelash,

We’re pleased to inform you that your manuscript has been judged scientifically suitable for publication and will be formally accepted for publication once it meets all outstanding technical requirements.

Kind regards,

Dereje Oljira Donacho, PhD

Academic Editor

PLOS One

Additional Editor Comments (optional):

Reviewers' comments:

Reviewer's Responses to Questions

**Comments to the Author**

Reviewer #1: All comments have been addressed

Reviewer #2: (No Response)

Reviewer #3: All comments have been addressed

2. Is the manuscript technically sound, and do the data support the conclusions?

Reviewer #1: Yes

Reviewer #2: (No Response)

Reviewer #3: Yes

3. Has the statistical analysis been performed appropriately and rigorously?

Reviewer #1: Yes

Reviewer #2: (No Response)

Reviewer #3: Yes

4. Have the authors made all data underlying the findings in their manuscript fully available?

Reviewer #1: Yes

Reviewer #2: (No Response)

Reviewer #3: Yes

5. Is the manuscript presented in an intelligible fashion and written in standard English?

Reviewer #1: Yes

Reviewer #2: (No Response)

Reviewer #3: Yes

Reviewer #1: The authors have addressed my comments adequately. The statistical methodology is more clearly described.

Reviewer #2: (No Response)

Reviewer #3: Dear Authors,

Greetings.

All comments were addressed to the article titled "Impact of Improved Stove Intervention on Infant Acute Respiratory Infections: Results from a Randomized Trial in Northwest Ethiopia" submitted to the Plos One journal.

Thank you very much.

Reviewer

**Do you want your identity to be public for this peer review?** For information about this choice, including consent withdrawal, please see our Privacy Policy

Reviewer #1: No

Reviewer #2: **Yes: ** Ayaan Hossain

Reviewer #3: **Yes: ** Beisan A. Mohammad

---

## [Editor Report · Acceptance letter]

PONE-D-25-24436R1

PLOS One

Dear Dr. Enyew,

I'm pleased to inform you that your manuscript has been deemed suitable for publication in PLOS One. Congratulations! Your manuscript is now being handed over to our production team.

Kind regards,

on behalf of

Dr. Dereje Oljira Donacho

Academic Editor

PLOS One